# Plasma Endothelial and Oxidative Stress Biomarkers Associated with Late Mortality in Hospitalized COVID-19 Patients

**DOI:** 10.3390/jcm11143950

**Published:** 2022-07-07

**Authors:** Arturo Orea-Tejada, Carlos Sánchez-Moreno, Octavio Gamaliel Aztatzi-Aguilar, Martha Patricia Sierra-Vargas, Dulce González-Islas, Yazmín Debray-García, Manolo Sibael Ortega-Romero, Candace Keirns-Davis, Laura Cornejo-Cornejo, Jorge Aguilar-Meza

**Affiliations:** 1Heart Failure and Respiratory Distress Clinic, Cardiology Service, Instituto Nacional de Enfermedades Respiratorias “Ismael Cosío Villegas”, Ciudad de México 14080, Mexico; oreatart@gmail.com (A.O.-T.); carlosmig.sanchez@gmail.com (C.S.-M.); mieshe@comcast.net (C.K.-D.); laucornejoc@gmail.com (L.C.-C.); jlam23go@gmail.com (J.A.-M.); 2Department of Toxicology and Environmental Medicine Research, Instituto Nacional de Enfermedades Respiratorias “Ismael Cosío Villegas”, Ciudad de México 14080, Mexico; gammaztatzi@gmail.com (O.G.A.-A.); yazmindebrayg@gmail.com (Y.D.-G.); rom_0@hotmail.com (M.S.O.-R.); 3Subdivision of Clinical Research, Instituto Nacional de Enfermedades Respiratorias “Ismael Cosío Villegas”, Ciudad de México 14080, Mexico; subinvestclinica@gmail.com

**Keywords:** COVID-19, oxidative stress, endothelial dysfunction, mortality

## Abstract

Background: Coronavirus infectious disease 2019 (COVID-19) is a significant public health problem worldwide. COVID-19 increases the risk of non-pulmonary complications such as acute myocardial injury, renal failure, thromboembolic events, and multi-organic damage. Several studies have documented increased inflammation molecules, endothelial dysfunction biomarkers, and dysregulation of coagulation factors in COVID-19 patients. In addition, endothelium dysfunction is exacerbated by the oxidative stress (OxS) promoted by endocrine and cardiovascular molecules. Our objective was to evaluate whether endothelial and OxS biomarkers were associated with mortality in hospitalized COVID-19 patients. Methods: A prospective cohort study was performed. Patients ≥18 years old with confirmed COVID-19 that required hospitalization were included in a prospective cohort study. Endothelium and oxidative stress biomarkers were collected between 3 and 5 days after admission. Results: A total of 165 patients were evaluated; 56 patients succumbed. The median follow-up was 71 days [23–129]. Regarding endothelial dysfunction and OxS biomarkers, patients who did not survive had higher levels of nitrates (0.4564 [0.1817–0.6761] vs. 0.2817 [0.0517–0.5], *p* = 0.014), total nitrates (0.0507 [−0.0342–0.1809] vs. −0.0041 [−0.0887–0.0909], *p* = 0.016), sE-Selectin (1.095 [0.86–1.495] vs. 0.94 [0.71–1.19], *p* = 0.004), and malondialdehyde (MDA) (0.50 [0.26–0.72] vs. 0.36 [0.23–0.52], *p* = 0.010) compared to patients who survived. Endothelial and OxS biomarkers independently associated with mortality were sE-selectin (HR:2.54, CI95%; from 1.11 to 5.81, *p* = 0.027), nitrates (HR:4.92, CI95%; from 1.23 to 19.63, *p* = 0.024), and MDA (HR: 3.05, CI95%; from 1.14 to 8.15, *p* = 0.025). Conclusions: Endothelial dysfunction (sE-selectin and nitrates) and OxS (MDA) are independent indicators of a worse prognosis in COVID-19 patients requiring hospitalization.

## 1. Introduction

Coronavirus infectious disease 2019 (COVID-19) is caused by the virus SARS-CoV2. It has become a public health problem worldwide, with significant transmissibility, increasingly high mortality, and the collapse of public health services [1,2].

SARS-CoV2 infection can occur as an asymptomatic disease or present with manifestations such as cough, shortness of breath, fever, pneumonia, pulmonary edema, severe acute respiratory syndrome (SARS), neurological and gastrointestinal symptoms in some cases, and even death [1,3].

The pathogenesis of the virus depends on the expression of angiotensin-converting enzyme 2 (ACE2) and kidney injury molecule-1, cell surface receptors that are present in higher proportions in the lung, kidney, and small intestine [4,5].

SARS-CoV 2 can infect the cells of the cardiovascular, respiratory, nervous, renal, and gastrointestinal systems due to the high expression of ACE2 receptors and transmembrane protease, serine 2 (TMPRSS2) that can be found in organs such as the esophagus, liver, and colon [6].

An altered intestinal microbiome combined with inflammation-increased intestinal permeability that allows toxin and bacterial translocation to the systemic circulation can contribute to multiorgan dysfunction [7].

COVID-19 increases the risk of non-pulmonary complications such as acute myocardial injury, renal failure, thromboembolic events, and multiorgan damage [8,9]. The common denominator of these phenomena may be a severe endothelial injury with subsequent dysfunction [10,11].

Endothelial dysfunction can cause a lower expression of vasodilatory and antithrombic molecules (e.g., nitric oxide (NO)). In the case of patients with COVID-19, this manifests as pulmonary vasculopathy, microangiopathy, thrombosis, and alveolar-capillary occlusion [12].

Endothelial dysfunction biomarkers include proteases, cellular adhesion molecules, glycocalyx components, coagulation factors such as tissue factor, sE-Selectin, endothelin-1, endogenous nitrite, nitrates, total nitrates, arginase, and plasminogen activator inhibitor type 1. Furthermore, an increased release of integrins and selectins during inflammation has been associated with higher endothelial activation; these biomarkers could be used as benchmarks for developing sepsis [13].

Several studies have documented increased inflammation molecules, endothelial dysfunction biomarkers, and dysregulation of coagulation factors, such as high levels of soluble endoglin and vascular cell adhesion molecule-1 (sVCAM1), in patients who did not survive COVID-19 [14,15]. Moreover, endothelial dysfunction is exacerbated by the oxidative stress (OxS) promoted by endocrine and cardiovascular molecules such as angiotensin-II, and it generates cell disequilibrium in the redox balance. OxS leads to the loss of biochemical properties of macromolecules that allow the development of lipoperoxidation (e.g., malondialdehyde (MDA)), protein carbonylation (e.g., advanced oxidized proteins products), glucose oxidized products (e.g., methylglyoxal (MGO), a precursor of advanced glycation products), and DNA oxidation (e.g., 8-oxoguanine). OxS biomarkers are related to the dysfunction of cardiovascular and respiratory systems, among others [16,17].

Our objective was to evaluate whether endothelial and OxS biomarkers were associated with mortality in hospitalized COVID-19 patients.

## 2. Materials and Methods

A prospective cohort study was performed at the Instituto Nacional de Enfermedades Respiratorias “Ismael Cosío Villegas” in Mexico City Mexico from 1 August 2019 to 31 March 2020.

Patients ≥ 18 years old with confirmed PCR tests for COVID-19 required hospitalization were included.

Demographic (age, sex), clinical variables (comorbidities, invasive mechanical ventilation use, PaO_2_/FiO_2_, organ failure assessment (SOFA) score, and biomarkers data) were noted, and endothelium and oxidative stress biomarkers were collected 3–5 days after admission.

The study was conducted according to the Declaration of Helsinki. It was approved by the Institutional Ethics and Research Committee of Biomedical Research in Humans of the Instituto Nacional de Enfermedades Respiratorias “Ismael Cosío Villegas” (approval number E-06-20).

Blood samples were centrifuged to separate the plasma fraction, and it was frozen at −80 °C until the measurement of the biomarkers. All samples were thawed and immediately analyzed for a total of one freeze-thaw cycle before use.

### 2.1. Endothelium and Oxidative Stress Biomarkers

NO^•^ was analyzed with the ELISA kit (KGE001, R&D Systems, Inc., Minneapolis, MN, USA). This method identifies the formation of nitrite at 540 nm from the enzymatic conversion of nitrate to nitrite by the enzyme nitrate reductase. A standard curve was constructed from 3.13 to 200 μmol/L by four-parameter logistic (4-PL) curve-fit. The minimum detectable dose (MDD) was 0.25 μmol/L. The results were expressed in μmol/L

Human sE-Selectin/CD62E (Cat. SSLE00) and Endothelin-1 (SET100) levels were measured using a human enzyme-linked immunosorbent assay (ELISA) kit (R&D Systems, Inc., Minneapolis, MN, USA). The measurements were performed following the specifications of each kit. E-Selectin concentrations were expressed in ng/mL.

MDA concentration. An aliquot of 100 μL of plasma was added to 650 μL of a solution of 1-methyl-2 phenylindole in a mixture of acetonitrile/methanol (3:1). The final concentration of the reagent was ten mM. The reaction was then initiated by adding 150 μL of 37% hydrochloric acid. The mixture was incubated at 45 °C for 40 min, and the reaction was measured at 586 nm [18].

Myeloperoxidase (MPO). Peroxidase activity of MPO (E.C. 1.11.2.2) with 3,3′,5,5′-Tetramethyl-benzidine dihydrochloride (TMB, Sigma, St. Louis, MO, USA) was measured as follows: a 10 μL plasma sample was combined with 50 μL of TMB solution (16 mM TMB in 14.5% DMSO [Sigma]), 100 mM sodium phosphate buffer at pH 5.4, and 17 μL 8.8 mM H_2_O_2_ (Sigma). The samples were incubated at 37 °C for 5 min. The reaction was stopped by adding 500 μL 0.4 M cold acetate buffer pH 3, and absorption was measured at 655 nm to estimate MPO activity [19]. One unit of MPO activity was defined as a 0.1 change in absorbance. MPO concentrations were expressed in U/mg of protein.

Methylglyoxal. Methylglyoxal (MGO) is a precursor of advanced glycated end-products able to interact with proteins by the Maillard reaction. The colorimetric MGO determination was performed with the 2, 4-dinitrophenylhydrazine (DNPH) alpha-keto acid method reported by Kwok et al. [20] and coupled to a microplate assay. Briefly, plasma samples (20 μL) were plated and mixed with 100 μL of DNPH (0.9 mM in 1N HCl) and incubated at 37 °C for 10 min. After that, the reaction was stopped by adding 100 μL of NaOH (1.5 N). The colored product was read at 540 nm in a microplate reader (LabSystems Multiskan MS microplate reader spectrophotometer). To calculate MGO, a concentration standard curve was established that followed the same procedure as the samples and used a 12.5 mM MGO solution as the standard (Cat. No M0252, Sigma Aldrich).

Arginase. Arginase (EC 3.53.1) catalyzes the hydrolysis of L-arginine to L-ornithine and urea. The microplating method was adapted from Corraliza et al. [21]. Briefly, plasma samples were diluted 1:10 with double distilled water and 100 μL phosphate buffer saline (0.1 M, pH 7.4; BioWhittaker Lonza Cat. No. 17517Q). The sample was incubated at 55 °C for 10 min. Aliquots of 50 μL of L-Arginine were added to reach a final concentration of 0.25 M in a final volume of 100 μL, then incubated for 1 h at 37 °C. After that, 150 μL of a reaction mix was prepared in a proportion of 1:15, composed of 9% isonitrosopropiophenone dissolved in ethanol and acid mix (H_2_SO_4_, H_3_PO_4_, H_2_O; 1:3:7 *v*/*v*). The final reaction was heated at 100 °C for 45 min. Before readings at 540 nm, samples were placed in ice and stood upright for 10 min in darkness. Urea was used as an internal control to construct a standard curve. Arginase concentrations were expressed as mg of Urea/mg of total protein.

Glutathione-S-Transferases (GSTs). The activity of GST (EC 2.5.1.18) was measured according to the method of Habig et al. [22] using chlorodinitrobenzene (CDNB) (Aldrich, Steinheim, Germany) as the substrate and reduced glutathione (Sigma). GST-CDNB conjugate formation was monitored by the change in absorbance at 340 nm. GSTs concentrations were expressed in nmol/min/mg protein.

Total plasma protein. Protein concentration was measured according to the method of Lowry et al. [23], and absorbance was determined at 550 nm using bovine serum albumin (BSA, Sigma) as a standard.

### 2.2. Statistical Analysis

Analyses were performed using a commercially available STATA version 14 (Stata Corp., College Station, TX, USA). Categorical variables were presented as frequencies and percentages. The Shapiro-Wilk test was used to test the normality of continuous variables. Continuous variables with normal distribution were presented as mean and standard deviation, and non-normal variables were presented as median and percentiles 25–75. A comparison among study groups was analyzed with a chi-square test or Fisher’s F tests for categorical variables and unpaired Student’s *t*-test or Mann-Whitney U tests for continuous variables. Kaplan-Meier survival curves and Cox’s proportional hazards analysis were performed to evaluate whether endothelial and OxS biomarkers were associated with mortality. We ran three Cox models, crude model, and adjusted models. Model 1 included age, sex, PaO_2_/FiO_2_, and invasive mechanical ventilation; model 2 included age, sex, PaO_2_/FiO_2_ and invasive mechanical ventilation, D-dimers, neutrophil-lymphocyte ratio, CRP, LDH, platelets, hemoglobin. A *p* < 0.05 was considered statistically significant.

## 3. Results

A total of 165 patients were evaluated, 56 of whom did not survive. The median follow-up during hospitalization was 71 days [23–129]. In survival patients, the median was 116 days [54–146], and in non-survival patients, it was 24.5 days [17.5–35]. The mean age of the population was 57.18 ± 13.37 years, 73.94% were men, and 84.24% required invasive mechanical ventilation.

Non-surviving subjects were older, with a higher frequency of invasive mechanical ventilation, lower PaO_2_/FiO_2_, and higher SOFA score than surviving subjects. Subjects who did not survive also had higher levels of inflammatory components-procalcitonin, D-Dimer, and CRP than surviving subjects. In addition, patients who succumbed had lower levels of lymphocytes, platelets, hemoglobin, hematocrit, total proteins, albumin, and globulin AG ratio, as well as higher neutrophil counts, direct bilirubin, and LDH than patients who survived (Table 1). Regarding endothelial dysfunction biomarkers, non-survivors had higher levels of nitrates (0.4564 [0.1817–0.6761] vs. 0.2817 [0.0517–0.5], *p* = 0.014), total nitrates (0.0507 [−0.0342–0.1809] vs. −0.0041 [−0.0887–0.0909], *p* = 0.016), and sE-Selectin (1.095 [0.86–1.495] vs. 0.94 [0.71–1.19], *p* = 0.004) compared to survivors (Figure 1). With respect to OxS biomarkers, non-survivors had higher levels of MDA (0.50 [0.26–0.72] vs. 0.36 [0.23–0.52], *p* = 0.010) compared to survivors (Figure 2).

Figure 3 and Figure 4 show the Kaplan-Meier curves for endothelial and oxidative stress biomarkers.

In the crude model, endothelial biomarkers associated with mortality were sE-selectin (HR: 1.21, CI 95%; from 1.02 to 1.44, *p* = 0.021), nitrates (HR: 1.83, CI95%; from 1.16 to 2.90, *p* = 0.009), total nitrates (HR: 2.53, CI 95%; from 1.15 to 5.55, *p* = 0.020), and arginase activity (HR: 44.79, CI 95%; from 1.96 to 1021.73, *p* = 0.017). In model 1, endothelial biomarkers associated with mortality were nitrates (HR: 1.98, CI 95%; from 1.24 to 3.16, *p* = 0.004), and total nitrates (HR: 2.78, CI 95%; from 1.22 to 6.32, *p* = 0.014). In model 2, sE-selectin (HR: 2.54, CI 95%; from 1.11 to 5.81, *p* = 0.027), nitrates (HR: 4.92, CI 95%; from 1.23 to 19.63, *p* = 0.024) (Table 2).

The OxS biomarker associated with mortality in the crude model was MDA (HR:3.16, CI 95%; from 1.57 to 6.39, *p* = 0.001); in model 1, MDA (HR: 3.90, CI 95%; from 1.75 to 8.66, *p* = 0.001); in model 2, MDA (HR:3.05, CI 95%; from 1.14 to 8.15, *p* = 0.025) (Table 2).

## 4. Discussion

The results of our study suggest that endothelial dysfunction and OxS affect the prognosis of SARS-CoV-2 infection. The virus spreads easily through several organs, and one of the most important determining factors of outcome is an endothelial function [6].

Once the virus invades various organs and affects type II pneumocytes, endothelial cells, smooth muscle, and macrophages: it leads to disseminated inflammation by overproduction of pro-inflammatory cytokines, macrophage recruitment, and pro-inflammatory granulocytes, producing a cytokine storm, endothelial dysfunction, microthrombi, small pulmonary vessel obstruction, vascular tone modification, and thrombosis in several vascular territories [24].

COVID-19 can disrupt the endothelial system, causing a massive release of von Willebrand factor, favoring thrombosis. Its propagation is facilitated by inflammation and endothelial dysfunction, which release Il-6 as a response to the virus, resulting in an amplified host immune response and cytokine storm [25]. With endothelial dysfunction, the expression of vasodilatory and antithrombic molecules is diminished; this is one of the characteristics of SARS-CoV2. In addition, generalized vasculopathy, microangiopathy, thrombosis, and alveolar-capillary occlusion have been found in the lungs of COVID-19 patients [12].

COVID-19-associated coagulopathy is the result of an endotheliopathy characterized by augmented von Willebrand factor release, platelet activation, and hyper-coagulability, leading to the clinical prothrombotic manifestations of COVID-19 that can include venous, arterial, and microvascular thrombosis. The factors responsible for this endotheliopathy and platelet activation are unclear. However, they might be explained by direct viral infection of endothelial cells, collateral damage to the tissue due to immune infiltration and activation, complement activation, or any number of inflammatory cytokines believed to play a role in COVID-19 disease [12,26,27].

The endotheliopathy in our cases was demonstrated by increased levels of OxS serum markers such as MDA generated from polyunsaturated fatty acid peroxidation in membrane cells [28,29]. We found that higher levels of serum MDA were independently associated with mortality (HR:3.05, CI 95%; 1.14 to 18.15, *p* = 0.025). A similar effect has been shown in other populations; higher serum MDA concentrations were independently associated with poor prognosis in patients with chronic heart failure and reduced ejection fraction [30]. The same was true in patients with sepsis, where higher MDA levels were associated with greater severity and mortality [31].

Endothelial dysfunction is characterized by the altered function of nitric oxide synthase and the bioavailability of NO. Both are evaluated in vivo with serum NO metabolites; nitrite and nitrate. Patients who recovered from COVID-19 showed lower nitrite and nitrite/nitrate ratio and higher nitrate levels than uninfected subjects [32]. COVID-19 causes SARS, where nitrate+nitrite concentrations were higher in patients with SARS who died, as reported by Sittipunt et al. [33]. Patients with acute hantavirus infection and non-infectious diseases [34,35,36] have shown higher nitrate levels [37]. Moreover, a higher plasma nitrate concentration has also been associated with a higher risk for all-cause mortality in the Framingham Offspring Study [38].

In our study, elevated levels of arginase activity were associated with mortality in hospitalized COVID-19 patients in the univariate model but not in the multivariate model. Increases in arginase activity have been linked to dysfunction and pathologies of the cardiovascular system, kidney, central nervous system, immune system dysfunction, and cancer [39]. Plasma arginase activity is even increased in diabetic subjects with altered NO synthase activity [40]. Arginase activity inhibitors may provide new therapeutic tools for vascular disease [41].

COVID-19 is also associated with hyperviscosity. According to Cheryl et al., in 15 patients diagnosed with COVID-19 pneumonia and admitted to the intensive care unit, all exceeded 95% of plasma viscosity, and this hyperviscosity was correlated with a high sequential SOFA score [42]. Hyperviscosity is associated with thrombosis, endothelial damage, and dysfunction [25,42]. During hospitalization, those patients with elevated serum levels of endothelial dysfunction markers had worse outcomes. However, this negative effect of SARS-CoV-2 infection is not limited to the critical stage. During post-COVID-19 follow-up, endothelial dysfunction probably continued to lead to increased mortality.

Endothelial dysfunction of any physiopathogenic mechanism may play a crucial role in chronic multiple organ damage, and any intervention to resolve it as soon as possible could improve the prognosis of these patients.

On the other hand, in our study, sE-Selectin was independently associated with a greater risk of death in hospitalized COVID-19 patients (HR:2.54, CI 95%; from 1.11 to 5.81, *p* = 0.027). sE-selectin is an inducible cell adhesion molecule expressed on endothelial cell surfaces exposed to shear stress. It plays an essential role in the recruitment of leucocytes into inflammation zones [36]. It is also considered a vasoactive substance marker of endothelial injury [43]. Significantly increased levels of serum sE-Selectin have been found in pathologies such as pulmonary hypertension [43] and sickle cell disease [44], among others [45,46,47]. It suggests that endothelial dysfunction and adhesion molecule expression contribute to disease pathogenesis. sE-Selectin has also been used as a marker of treatment response in patients with rheumatoid arthritis treated with immunosuppressors [48] and burn patients treated with atorvastatin [49].

In addition, the neutrophil-lymphocyte ratio has been considered a marker of systemic inflammation in COVID-19 subjects and other populations [50,51]. An elevated neutrophil-lymphocyte ratio is associated with a worse prognosis. Similar results were found in our study; subjects who did not survive had higher of neutrophil-lymphocyte ratios than subjects who survived.

On the other hand, comorbidities such as cancer, chronic kidney diseases, diabetes mellitus, and hypertension have been associated with mortality in COVID-19 patients [52]. However, in our study, we did not find differences between the study groups; this could be due to the small sample size of our research.

## 5. Strength and Limitations of the Study

Among the study’s limitations is the small sample size; however, its main strength is that it is a prospective cohort study, which allowed us to examine the impact of endothelial function and OxS on the prognosis of patients hospitalized for COVID-19.

## 6. Conclusions

The SARS-CoV2 infection causes endothelial dysfunction and OxS by several mechanisms in the endothelial cells. Identification of endothelial dysfunction markers (sE-selectin and nitrates) and OxS (MDA) are independent risk markers of a worse prognosis in COVID-19 patients requiring hospitalization. Finally, the early therapeutic strategies for this might improve long-term treatment results.

## Figures and Tables

**Figure 1 jcm-11-03950-f001:**
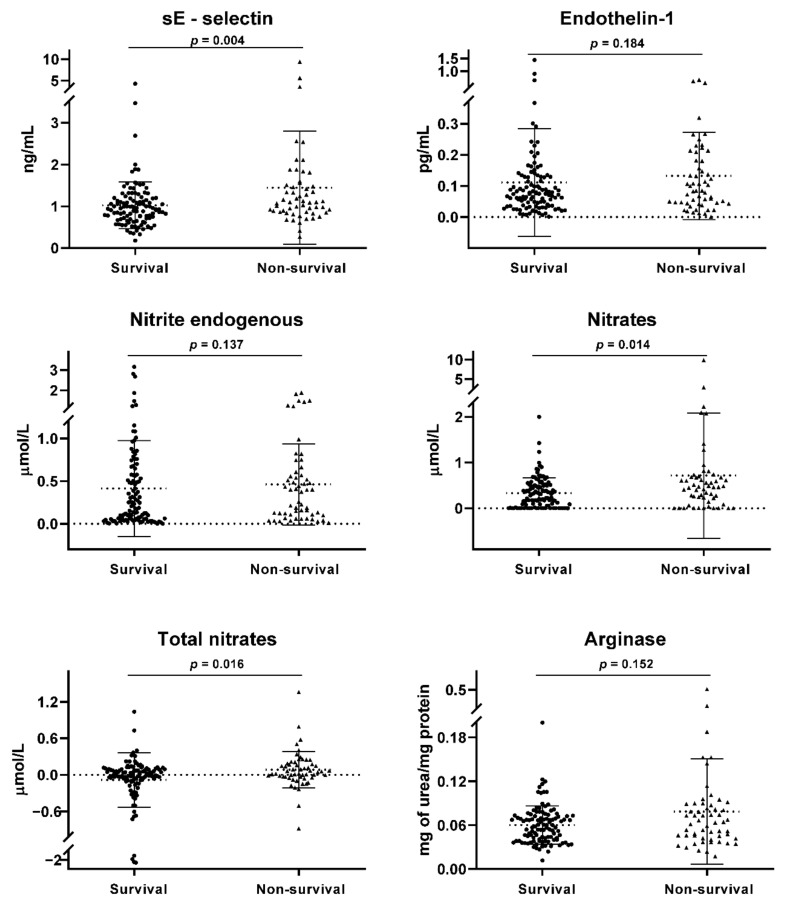
Endothelial dysfunction biomarkers. A comparison of medians between the study groups was performed using the Mann-Whitney test. A *p* < 0.05 was considered statistically significant.

**Figure 2 jcm-11-03950-f002:**
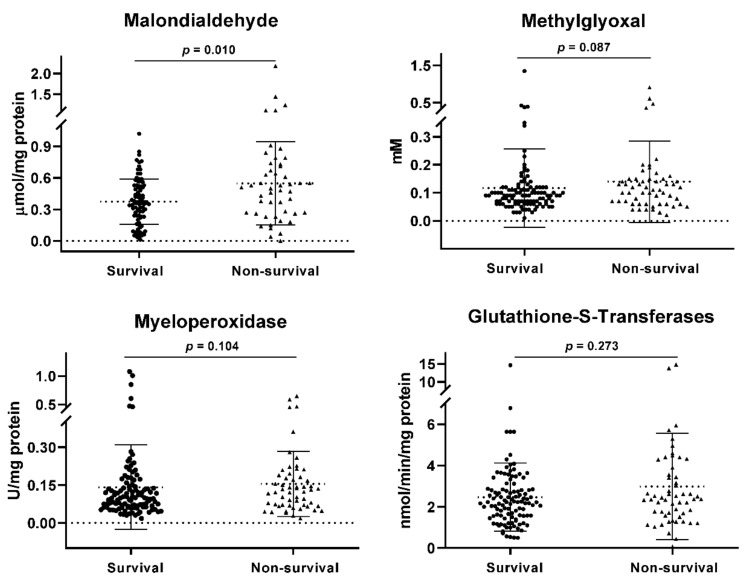
Oxidative stress biomarkers. A comparison of medians between the study groups was performed using the Mann-Whitney test. A *p* < 0.05 was considered statistically significant.

**Figure 3 jcm-11-03950-f003:**
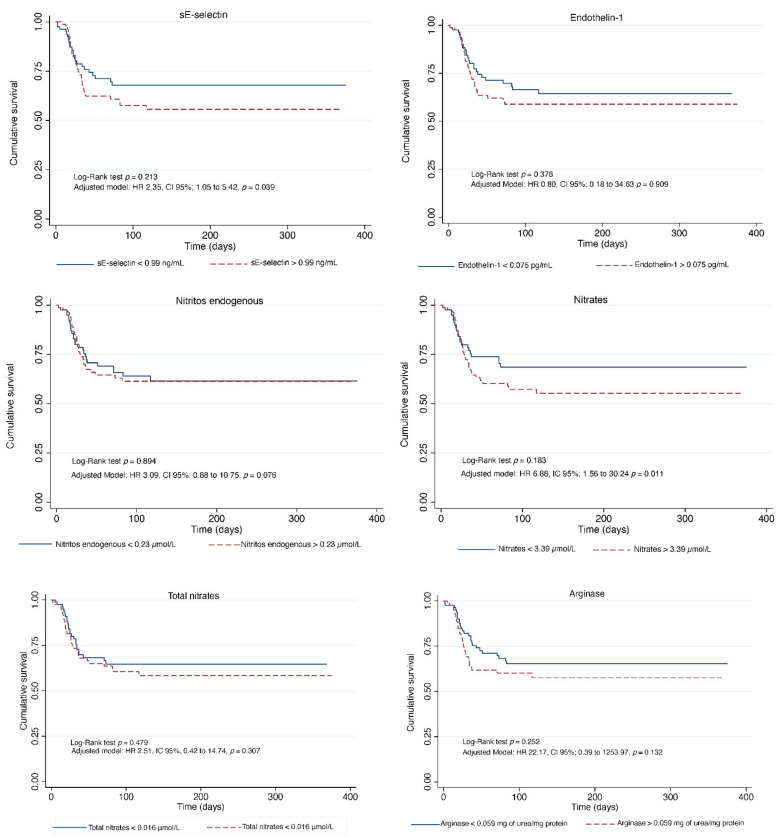
Kaplan-Meier curves for endothelial biomarkers.

**Figure 4 jcm-11-03950-f004:**
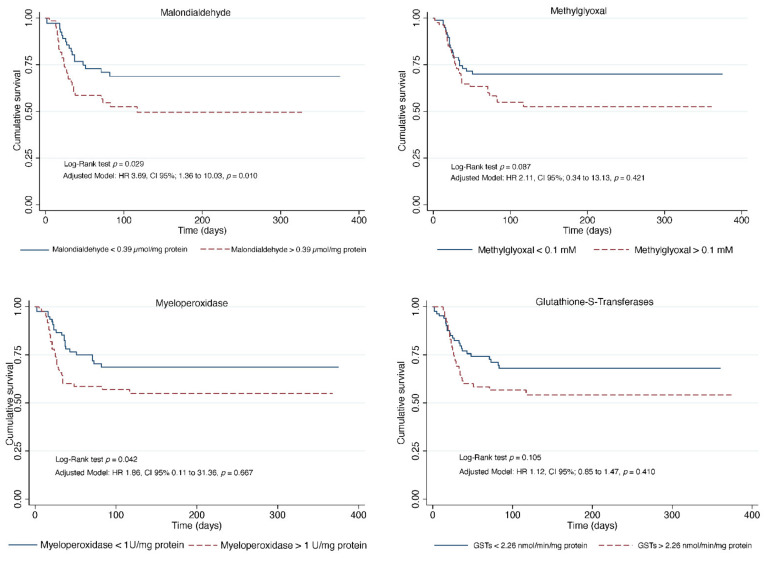
Kaplan-Meier curves for oxidative stress biomarkers.

**Table 1 jcm-11-03950-t001:** Baseline characteristics.

	All n = 165	Non-Survival n = 56	Survival n = 109	*p*-Value
Men, n (%)	122 (73.94)	46 (82.14)	76 (69.72)	0.085
Age, y	57.18 ± 13.37	62.71 ± 13.52	54.34 ± 12.43	<0.001
**Comorbidities**				
Hypertension, n (%)	59 (35.76)	19 (33.93)	40 (36.70)	0.725
Diabetes, n (%)	51 (30.91)	22 (39.29)	29 (26.61)	0.095
Obesity, n (%)	65 (41.67)	21 (40.38)	44 (42.31)	0.818
Cardiovascular disease, n (%)	11 (6.67)	5 (8.93)	6 (5.50)	0.512
Chronic kidney disease, n (%)	4 (2.42)	2 (3.57)	2 (1.83)	0.605
**Ventilatory parameters**				
Invasive mechanical ventilation, n (%)	139 (84.24)	53 (94.64)	86 (78.90)	0.009
PaO_2_/FiO_2_, mmHg	144.46 ± 50.30	128.39 ± 49.14	153.07 ± 49.01	0.004
Oxygen saturation, %	70.5 [58–83]	70 [50–82]	71 [60–83.5]	0.290
SOFA score	6 [3–9]	8 [4–10]	6 [3–8]	0.001
**Biomarkers data**				
PT, s	14.5 [13.8–15.7]	14.75 [14–16.6]	14.4 [13.7–15.6]	0.095
INR	1.02 [0.97–1.10]	1.02 [0.97–1.16]	1.01 [0.96–1.10]	0.171
APTT, s	42.4 [38.10–49.5]	42.15 [38.1–51.9]	42.4 [37.6–48.8]	0.538
Procalcitonin, ng/ml	0.26 [0.09–0.98]	0.54 [0.25–4.48]	0.17 [0.07–0.56]	<0.001
D-dimers, μg/ml	1.53 [0.73–2.83]	1.75 [0.83–4.14]	1.4 [0.66–2.72]	0.030
CRP, mg/L	7.90 [2.49–16.7]	11.25 [6.56–19.04]	5.03 [1.47–14.91]	0.001
Leukocytes, mm^3^	10.8 [7.5–13.7]	11.1 [8.65–14.2]	10.6 [6.7–13.7]	0.132
Neutrophils, %	8.5 [5.9–11.5]	8.5 [7.4–12.25]	8.5 [4.9–11.4]	0.043
Lymphocytes, %	0.9 [0.6–1.4]	0.7 [0.45–1.15]	1 [0.7–1.5]	<0.001
Neutrophil-lymphocyte ratio	9 [5.46–16.7]	12.89 [8.58–23.15]	7.6 [4–13.77]	< 0.001
Platelets, mm^3^	309 [223–416]	265 [161.5–378]	329 [256–422]	0.003
Hemoglobin, g/dL	12.04 ± 2.53	11.38 ± 2.54	12.38 ± 2.48	0.018
Hematocrit, %	35.63 ± 7.19	33.97 ± 7.21	36.48 ± 7.07	0.036
Total proteins, g/dL	5.70 ± 0.77	5.38 ± 0.81	5.86 ± 0.69	<0.001
Albumin, g/dL	2.69 ± 0.53	2.41 ± 0.50	2.83 ± 0.49	<0.001
Globulin, g/dL	3.0 ± 0.54	2.96 ± 0.55	3.02 ± 0.54	0.534
Globulin AG ratio, g/gL	0.86 [0.74–1.03]	0.77 [0.69–0.91]	0.90 [0.77–1.11]	<0.001
Total bilirubin, mg/dL	0.54 [0.43–0.78]	0.56 [0.43–0.89]	0.53 [0.44–0.75]	0.313
Direct bilirubin, mg/dL	0.16 [0.1–0.25]	0.21 [0.13–0.32]	0.14 [0.10–0.23]	0.006
Indirect bilirubin, mg/dL	0.37 [0.29–0.49]	0.34 [0.26–0.52]	0.38 [0.30–0.48]	0.610
γ-GT, U/L	122.5 [77–214]	106.5 [61–182.5]	127.5 [80–252]	0.100
LDH, U/L	329.5 [251.5–428]	378.5 [286.5–493.5]	296 [243–411]	0.007
CPK, U/L	88 [37–230]	107.5 [49.5–317]	61 [31–204]	0.053

PT—prothrombin time; INR—international normalized ratio; APTT—activated partial thromboplastin time; CRP—C-reactive protein; γ-GT—γ-Glutamyl transpeptidase; LDH—lactate dehydrogenase; CPK—creatine phosphokinase.

**Table 2 jcm-11-03950-t002:** Plasma endothelial and oxidative stress biomarkers associated with mortality in hospitalized COVID-19 patients.

	Crude Model			Model 1			Model 2		
	HR	CI 95%	*p*	HR	CI 95%	*p*	HR	CI 95%	*p*
**Endothelial biomarkers**									
sE-selectin, ng/mL	1.21	1.02–1.44	0.021	1.16	0.97–1.39	0.084	2.54	1.11–5.81	0.027
Endothelin-1, pg/mL	1.40	0.37–5.23	0.610	0.86	0.22–3.33	0.830	1.38	0.023–75.63	0.870
Nitrites endogenous, μmol/L	1.07	0.67–1.70	0.753	1.14	0.69–1.88	0.587	1.78	0.60–5.31	0.297
Nitrates, μmol/L	1.83	1.16–2.90	0.009	1.98	1.24–3.16	0.004	4.92	1.23–19.63	0.024
Total nitrates, μmol/L	2.53	1.15–5.55	0.020	2.78	1.22–6.32	0.014	2.51	0.42–14.74	0.307
Arginase, mg of urea/mg protein	44.79	1.96–1021.73	0.017	20.76	0.71–604.56	0.078	9.73	0.21–450.76	0.245
**Oxidative stress biomarkers**									
Malondialdehyde, μmol/mg protein	3.16	1.57–6.39	0.001	3.90	1.75–8.66	0.001	3.05	1.14–8.15	0.025
Methylglyoxal, mM	1.97	0.49–7.95	0.336	1.79	0.40–7.91	0.438	2.25	0.33–15.23	0.406
Myeloperoxidase, U/mg of protein	1.39	0.32–5.90	0.652	1.81	0.36–9.15	0.470	1.48	0.10–21.71	0.773
GSTs, nmol/min/mg protein.	1.08	0.98–1.19	0.093	1.04	0.95–1.15	0.343	1.16	0.87–1.54	0.301

Model 1 adjusted by age, sex, PaO_2_/FiO_2_, and invasive mechanical ventilation. Model 2 was adjusted by age, sex, PaO_2_/FiO_2_, invasive mechanical ventilation, D-dimers, neutrophil/lymphocyte ratio, CRP, LDH, platelets, and hemoglobin. GSTs—Glutathione-S-Transferases.

## Data Availability

The datasets generated and/or analyzed during the current study are not publicly available due to the fact that individual privacy could be compromised but are available from the corresponding author on reasonable request.

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
