# Peer review of "Plasma Endothelial and Oxidative Stress Biomarkers Associated with Late Mortality in Hospitalized COVID-19 Patients"

_jcm, 2022, doi:10.3390/jcm11143950_

Round 1

Reviewer 1 Report

Here, Orea-Tejada and colleagues performed a single-center prospective study to evaluate the prognostic value of biomarkers of endothelial dysfunction and oxidative stress in hospitalized patients with COVID-19. They showed that E-selectin, nitrates, arginase, and malondialdehyde are associated with in-hospital mortality.

Although the topic of the noninvasive assessment of endothelial dysfunction in critically ill patients is certainly worth of investigation, the manuscript presents some major flaws that prevent an unbiased evaluation of the authors' conclusions.

First, it is unclear when the biomarkers were assessed (at admission?). Moreover, the Cox regression performed to retrieve biomarkers associated with mortality does not seem to be adjusted for potential confounders, first of all, age - which was significantly different between groups - and sex, but also the presence of diabetes and other inflammatory biomarkers. Furthermore, it is unclear which statistical tests were performed for the comparisons in Fig. 1 and 2

Moreover, there seem to be some discrepancies between the results of the comparisons in Figure 1 and the regression shown in Table 2, which are very difficult to justify in absence of some adjustments to the Cox regression. For example, arginase activity was not significantly different between groups in Fig. 1 but significantly associated with mortality in Table 2.

The manuscript could have also benefited from a Kaplan-Meier analysis of each biomarker stratified in tertiles, or according to the median.

Finally, in the Discussion, the authors state that "Considering our findings it is warranted to speculate that endothelial dysfunction of any physiopathogenic mechanism plays a crucial role in chronic multiple organ damage ". To support this conclusion, the authors should have performed at least some kind of regression analysis to find correlations between the assessed biomarkers and markers of organ dysfunction, such as eGFR, PaO2/FiO2, and arterial pH, among others.

Reviewer 2 Report

In the paper of Orea-Tejada and colleagues, the authors aim to test the impact of oxidative stress and endothelial biomarkers, along with other variables, on the prognosis of COVID-19 patients. They found several interesting results supporting the hypothesis that endothelial disjunction and oxidative stress play a role in the COVID-19 harmful mechanisms. 

Overall the paper is well written, and the results are in line with the publication in the JCM. I have some concerns listed below.

1 Abstract

In the Methods section of the abstract, the authors explain the aim, not the methods. Please provide a section explaining the sample selection and included variables. In addition, in the abstract, there is no conclusion section.

2 Introduction

The sentence "In a Meta-analysis, Dalla-Sega y Cols. Shown a mortality rate of 5.5%, which is higher than the informed by Cambridge Crystallographic Data Centre, 2.38%, due to most of the patients were hospitalized and 6.1% of them were in a critical condit" is odd. Please provide further explanation or remove it.

3 Methods

I would suggest the authors specify all the variables they considered in the analysis in the methods section, and not only in the table. These include comorbidities and ventilatory scores. Please also specify all the used scales and scores, including SOFA, that the author mentioned in the results section without defining it earlier.

Malondialdehyde has already been abbreviated to MDA in the introduction section.

4 Results

Please specify the meaning of follow-up in this study and whether it refers to a post-hospitalization follow-up or the hospitalization period. In addition, the authors report the median follow-up of the whole population, it would be interesting to search for differences in follow-up lenght between survivors and non-survivors.

Please define "total nitrates", which appears in the methods section, in the figure 1 but not in table 2 (please add if the variable total nitrates has been included in the analysis)

5 Discussion

Please define the abbreviation VWF.

Considering the sample size, I would suggest editing the sentence "The results of our study demonstrate that endothelial dysfunction and OxS have an effect on the prognosis of SARS-CoV-2 infection" in order to make it less assertive.

Since the authors found significant differences between survivors and non-survivors in the neutrophil and lymphocyte counts, it would be interesting to discuss this finding. Please also consider previous evidence regarding the prognostic role of the neutrophil to lymphocyte ratio in covid-19 patients (e.g., Predictive values of neutrophil-to-lymphocyte ratio on disease severity and mortality in COVID-19 patients: a systematic review and meta-analysis). It would be interesting to add it as a further analysis.

The authors found no significant differences between groups regarding comorbidities. Please discuss this lack of evidence (possibly related to the small sample), since it is well known that comorbidities affect prognosis in covid-19 patients ( e.g., Comorbidities in SARS-CoV-2 Patients: a Systematic Review and Meta-Analysis).

Round 2

Reviewer 1 Report

The authors addressed all of my concerns.

Author Response

The authors thank the reviewer for the valuable comments and time invested in our manuscript. 

Reviewer 2 Report

Minor concerns:

ABSTRACT

Please specify the abbreviation MDA at its first appearance (page 1, line 41)

The last sentence “Endothelial dysfunction (sE- 44 selectin and nitrates) and OxS (MDA) are and independent risk markers of a worse prognosis” should specify “in COVID-19 patients requiring hospitalization"

METHODS

Please specify the difference between nitrates and total nitrates (page 3, line 117)

RESULTS

Please specify in the caption of figure 1 and 2 the statistical threshold

Please underline statistically significant results in table 2 (specifying the statistical threshold in the caption)

Author Response

This manuscript is a resubmission of an earlier submission. The following is a list of the peer review reports and author responses from that submission.